# 180 years of marine animal diversity as perceived by public media in southern Brazil

Dannieli Firme Herbst[1]*, Jara Rampon[2], Bruna Baleeiro[2], Luiz Geraldo Silva[3], Thiago Fossile[1], André Carlo Colonese[1]*

1 Institute of Environmental Science and Technology (ICTA) and Department of Prehistory, Universitat Autònoma de Barcelona, Bellaterra, Spain, 2 Departament of Ecology and Zoology, ECZ/CCB, Universidade Federal de Santa Catarina, Campus Trindade, Florianópolis, SC, Brazil, 3 Department of History, Universidade Federal do Paraná (UFPR), Conselho Nacional de Desenvolvimento Científico e Tecnológico (CNPq), Brazil

* dannieli.firme@uab.cat (DFH); andrecarlo.colonese@uab.cat (ACC)

**Data Availability Statement:** All relevant data are within the paper and its Supporting Information files.

## Abstract

Commoditization of marine resources has dramatically increased anthropogenic footprints on coastal and ocean systems, but the scale of these impacts remain unclear due to a pervasive lack of historical baselines. Through the analysis of historical newspapers, this paper explores changes in marine animals (vertebrates and invertebrates) targeted by historical fisheries in southern Brazil since the late 19th century. The investigation of historical newspaper archives revealed unprecedented information on catch composition, and perceived social and economic importance of key species over decades, predating official national-level landing records. We show that several economically and culturally important species have been under persistent fishing pressure at least since the first national-scale subsidies were introduced for commercial fisheries in Brazil in the late 19th and early 20th centuries. Our work expands the current knowledge on historical fish catch compositions in the southwestern Atlantic Ocean, while advocating for the integration of historical data in ocean sustainability initiatives.

## Introduction

Coastal and ocean systems have played a crucial role in human evolution, providing food, health, well-being and livelihoods since prehistoric times [1,2]. Marine biodiversity, in particular, provides enhanced nutritional benefits to human societies [3], while ensuring fundamental ecosystem functions, structures and services [4,5]. Since the second half of the 18th century (Industrial Era), and especially in recent decades, pervasive and compounded anthropogenic impacts on ocean and coastal systems have accelerated marine biodiversity loss and population decline, and increased the vulnerability of marine ecosystem services [4,6–8]. Understanding the scale of anthropogenic stressors, however, is extremely complex and exacerbated by the paucity of long-term records on species composition, abundance and distribution pre-dating the most recent decades of anthropogenic impacts [9,10]. This form of historical amnesia creates misconceptions that have far-reaching consequences for sustainability and conservation

**Funding:** This work was funded by the ERC Consolidator project TRADITION, which has received funding from the European Research Council (ERC) under the European Union's Horizon 2020 research and innovation programme under Grant Agreement No 817911, awarded to ACC. This work was also funded by EarlyFoods (Evolution and impact of early food production systems), 00527 (Generalitat de Catalunya, SGR-Cat 2021), awarded to ACC. This work was also funded by the ICTA-UAB "María de Maeztu" Programme for Units of Excellence of the Spanish Ministry of Science and Innovation (CEX2019-000940-M). The funders had no role in study design, data collection and analysis, decision to publish, or preparation of the manuscript.

**Competing interests:** The authors have declared that no competing interests exist.

actions. For example, it may conceivably increase tolerance for progressive environmental degradation and contribute to setting inappropriate sustainability targets and responses by ocean stakeholders; and hence underpin their expectations as to what is a desirable and achievable state of social-ecological systems [11–13]. These are typical symptoms of the "shifting baseline syndrome" [14] that require locally-impactful and globally-relevant new knowledge to engage and support stakeholders in designing conservation pathways to ocean sustainability.

Moreover, the Intergovernmental Oceanographic Commission of the United Nations Educational, Scientific, and Cultural Organisation (IOC-UNESCO) has recently launched important global efforts to educate people on the oceans (Ocean Literacy), including fundamental concepts about how oceans function and human-ocean interdependencies [15,16]. Understanding humanity's past, present and future relationships with the oceans are therefore of utmost importance to both the on-going implementation of the UN Decade on Ecosystem Restoration (2021–2030; [17]) and the UN Decade on Ocean Science for Sustainable Development [15,16].

The southern Atlantic Forest coast of Brazil is a world biodiversity hotspot with a long history of human-ocean interaction, and a priority region for socially equitable sustainable development in cooperation with biological conservation and restoration [18,19]. It is also, however, a data poor region in relation to fisheries statistics despite being the largest fish producing region (Santa Catarina state) in the country [20]. Stock monitoring data are regionally and chronologically dispersed [20–23], and mostly limited to the second half of the 20th and the first decade of the 21st centuries [20,24–27]. In Brazil, there has been an increased effort to expand stock reconstructions with data collected from local fisheries [28–33] and seafood wholesale market data [34,35], but these approaches are inherently limited to only the most recent decades affected by large national fisheries subsidies, specifically from the 1960's to 1970's [36]. Overall, little is known about the diversity of targeted species and their socio-cultural, economic and market values before more recent political and financial support were introduced to the fishing industry.

Historical sources can provide valuable insights into past species composition, distribution and abundance [6,9,10,37–39]. Among written public media documents, newspapers offer some advantages for qualitative and quantitative historical analyses [40–43]. Newspapers generally have large distribution ranges and relatively high frequency of issues (released on a daily to semestral basis), while digitization programmes through public funds and advances in optical character recognition (OCR) technology are making historical newspapers increasingly accessible [43–46]. Moreover, public media offers insights into societal (e.g. cultural, economic and political) drivers of change, including stakeholder perception of environmental conditions through time [43,47]. Public media also shapes public opinion [48], influencing peoples' literacy of ocean, human and ecological affairs [16], and thus may be a major driver of social-ecological baselines in all societies. There is a rich body of literature conceptualising the use of newspapers for reconstructing historical events that can help marine historical ecologists assess data quality while framing their research in the evolving editorial and political agenda of popular media through time [49,50].

In Brazil, newspapers have been issued since the beginning of the 19th century, and are one of the few sources of information documenting the development of political and economic institutions, and commercial activities over the last centuries [49,51,52]. Gallardo et al. [43] have recently shown that historical newspapers retained insightful information on the drivers of anthropogenic impacts on coastal areas of Brazil over the last 150 years. Here we expand their approach to derive long-term data on marine organisms targeted by Brazil's fishing sector between 1840 and 2019, and notably during the first half of the 20th century, preceding "blue acceleration" of the last decades [8], where a major gap in understanding still exists. We

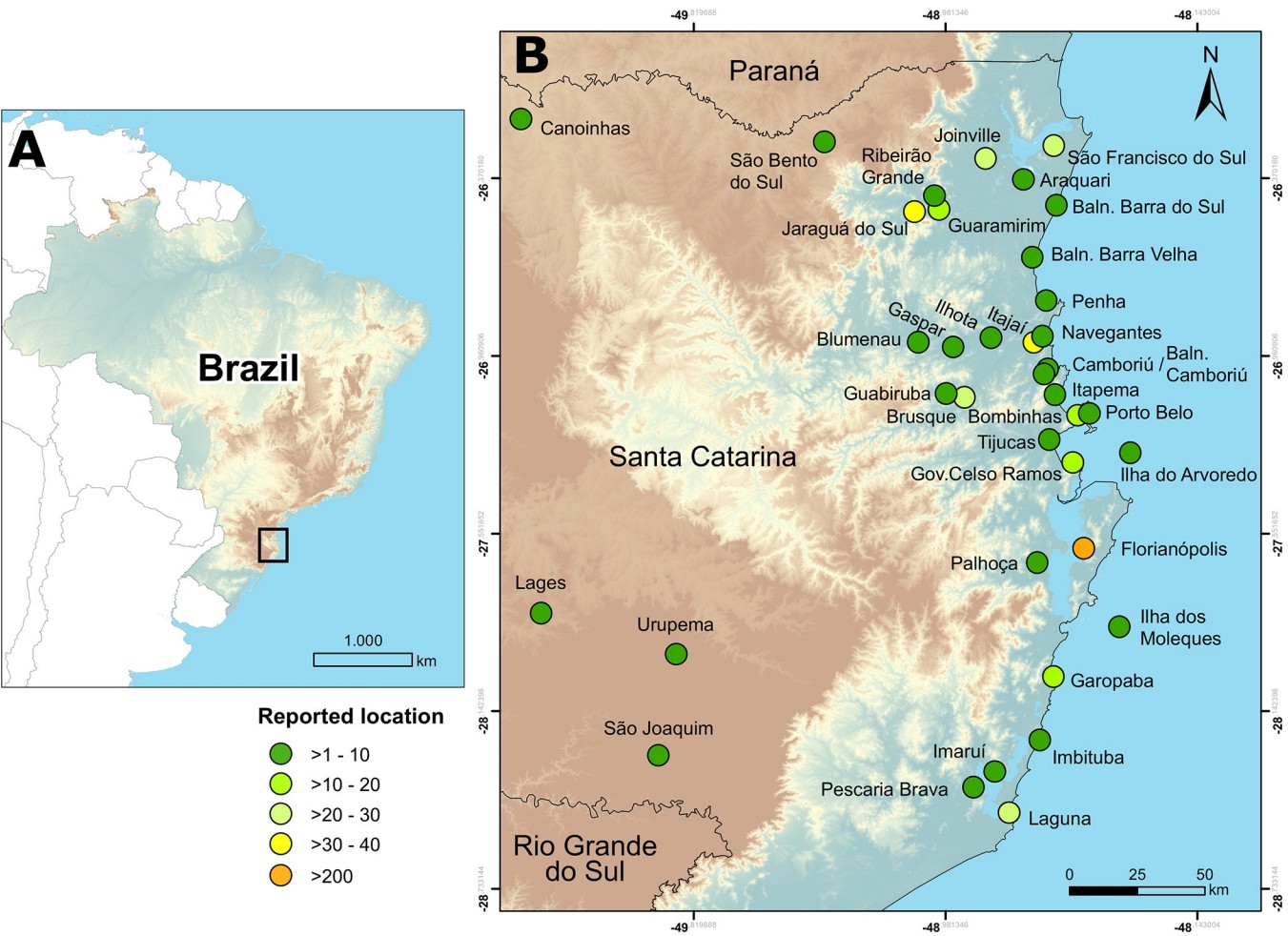

**Fig 1.** Map of the focal research area (A) and localities (B) in the state of Santa Catarina reported in the newspapers. The colour gradient of the localities represents the number of reports identified per municipality. The majority of the reported organisms derived from captures and landings in Florianopolis, the capital of Santa Catarina. Maps generated using ArcGIS 10.7 (https://desktop.arcgis.com/en/), Inkscape (https://inkscape.org/), and GNU Image Manipulation Program (https://www.gimp.org/), on data publicly available from Natural Earth (https://www.naturalearthdata.com/), National Institute for Space Research—INPE (http://terrabrasilis.dpi.inpe.br/; [53]) and CGIAR Consortium for Spatial Information (https://cgiarcsi.community/data/srtm-90m-digital-elevation-database-v4-1/; [54]).

investigated historical newspapers published in Santa Catarina state (Fig 1) and scrutinised more than twenty thousand digitised editions in the Brazilian Digital Newspapers and Periodicals Library (Hemeroteca Digital Brasileira). Our aim is to uncover the diversity of species commercially exploited, traded and consumed in southern Brazil since the late 19th century, prior to the start of the global-level blue acceleration social-ecological phenomena [8] and the first large fisheries subsidies in the country [36]. Our paper offers new insights into the origin and changing nature of significant human impacts on marine systems in the southwestern Atlantic Ocean; as well as implications for ocean sustainability initiatives aiming to bring society's social-ecological baselines closer to historical reality.

## Material and methods

### Data collection

The sampling method follows Gallardo et al. [43] and, in brief, it can be summarised in the following steps: A) Historical newspapers were sourced from the Brazilian Digital Newspapers

and Periodicals Library (*Hemeroteca Digital Brasileira*, *Fundação Biblioteca Nacional*, *BNDigital*), made available by the National Library Foundation (*Fundação Biblioteca Nacional*, https://bndigital.bn.gov.br/hemeroteca-digital/). B) News items (e.g. articles, opinion letters, regulations and sanctions) were sourced using the keywords "*pesca*" (fishing) and "*peixe*" (fish), selecting the "*Locality*" as the region of Santa Catarina using the State abbreviation "*SC*", opting for single decades as the "*Period*", and including all newspapers for the selected decades. Details of the searching facilities including OCR available through the *BNDigital* can be found at https://bndigital.bn.gov.br/hemeroteca-digital/. The keywords "*pesca*" (fishing) and "*peixe*" (fish) were chosen for the following reasons: i) The keyword "*pesca*" captured several types of marine resources, and also captured the word "*pescado*" which categorically addresses different types of resources exploited by fisheries; ii) within Brazilian folk taxonomy (ethnotaxonomy), various marine animals ("*pescado*") are classified as "*fish*" [55,56]; iii) the popular association of fish such as shrimp, lobster, oysters, octopus and squid with crustaceans and molluscs is something very recent and thus this nomenclature is infrequently used in everyday life; iv) given the large diversity of organisms captured by coastal fisheries in Brazil and the general scope of the paper (not targeting a specific species), it would be unfeasible to extend the keywords to all potential captured organisms. C) Single items were then assessed for duplicate results and non-relevant information (e.g. commercial advertisements) which were then excluded. Once the search was completed, single items were fully read in order to assess the content and quality of information based on two inclusion criteria: i) the items focused on coastal and ocean areas, including estuaries and coastal lagoons, or coastal rivers; and ii) the items reported marine animals (e.g. fish, crustaceans, molluscs, mammals, reptiles), including non-native taxa (S1 File). D) An analytical framework was created to collect and categorise content and contextual information, which consisted of 1) a catalogue of reported taxa and their overall trophic ecology, and 2) the type of fishing gear, when reported (SI1). Aquatic animals were mostly reported using vernacular names, with conversion to scientific names (using the minimum identified taxonomic level), whenever possible, based on local literature [57]. The catalogued animals and gear were quantified for their absolute and relative abundances (the number of times a species or type of gear occurred in a given number of newspaper items) and richness (the number of different species and gear types in a given number of newspaper items) aggregated per decade. If species *X* was reported more than once in a single item, its abundance in the given item was counted as one.

## Data analysis

For each species, the trophic level (TL) was assigned according to data in FishBase, using the main diet of adults [58]. For taxa where only genus and/or family were identified we used the average values of the main species in the region according to SiBBr (https://sibbr.gov.br). With this information we computed the weighted average trophic level (WATL) for each decade. In addition, we computed fish-based piscivores/planktivores (Pis/Pla) [59] and invertebrates/fish (Inv/Fis) to explore changes in overall catch composition through time. For this we used the absolute abundance of species reported in newspapers per decade. These are simple ecological indicators that are used here to infer changes in fishing pressure, trophic level (fishing down the food web [60]), biological productivity, market preferences and technological changes [34,59]. Non-native and freshwater organisms (e.g. fish, shrimp), reptiles, birds and sea mammals were excluded from WATL, Pis/Pla, Inv/Fis analyses. The results from Inv/Fis were compared with reconstructed and officially reported industrial and artisanal catch data (volume in tons) for Santa Catarina from 1950 to 2015 [20]. Piscivores included fish species consuming finfish and pelagic cephalopods, while for planktivores their diet consisted of zooplankton and

early stages of fish and jellyfish [59]. For these two trophic categories we used references from Quimbayo et al. [61], complemented with other literature [58,62,63]. When combined, multiple indicators are expected to provide a more comprehensive picture of historical changes in the system [34]. Fishing gear were standardised following the Fishing Gear Type of the ICM-Bio (*Centro Nacional de Pesquisa e Conservação da Biodiversidade Marinha do Sudeste e Sul* (https://www.icmbio.gov.br/cepsul/artes-de-pesca.html)). The information collected from the newspapers were complemented with additional sources, such as the reports of the Ministry of Agriculture (available at http://ddsnext.crl.edu/). Data collection was carried out from March 2021 to March 2022.

## Results

### Species composition and fishing technology

The keywords "*pesca*" (fishing) and "*peixe*" (fish) generated 9,190 and 14,168 matches respectively, spanning a period of 180 years (1840 to 2019). Over 23,400 newspaper reports (items) were individually assessed and read. Of these, 598 items were selected following the inclusion criteria reported above. The selected items were reported in 57 newspapers, representing 4.7% of the total newspapers available in the HBD for Santa Catarina over the studied time period (n = 1206, 18th March 2022). The selected items provided information for 37 municipalities, of which 49% (n = 298) were related to Florianópolis (formerly Desterro, Fig 1B), the capital of the state [43]. From 1950 onwards there was an increase in news items from municipalities located by the Itajaí River and the northern coastal regions of the state (e.g. Barra Velha and São Francisco do Sul), where Brazil's largest commercial fishing industry developed over the last decades.

The keywords "*pesca*" (fishing) and "*peixe*" (fish) captured animals from different taxonomic groups (fish, crustaceans, molluscs, mammals, reptiles), which is significant since in folk taxonomy several taxa are also considered as "fish". Native and non-native animals were mostly reported as vernacular (*v*) and, to a lesser extent, as scientific (*s*) names, the latter sporadically appearing in more recent decades. We found 1655 mentions of marine animals, totaling 277 and 203 vernacular and scientific taxa respectively (SI1). As reported by Feire and Pauly [64], we also found evidence that a single folk nomenclature was used for several species (e.g. *Hypanus* sp., *Dasyatis* sp., *Fontitrygon* sp., *Bathytoshia* sp. = *raia prego*), and several folk nomenclatures were also used for a single species (e.g. *Mugil liza = tainha*, *tainhota*, *tanhota*, *do corso*, *facão*, *tainha-assú*). As a consequence, attempts to correlate scientific and vernacular names were complex and not always possible with an equal level of confidence.

Native marine fish dominated in terms of number of species, both vernacular (*v* = 192) and scientific names (*s* = 143), as well as in abundance (n = 1,041). This was followed by crustaceans (*v* = 17, *s* = 9, n = 181), freshwater animals (*v* = 22, *s* = 15, n = 111), molluscs (*v* = 11, *s* = 9, n = 99), sea mammals (*v* = 9, *s* = 7, n = 81), echinoderms (*v* = 2, *s* = 2, n = 2), reptiles (*v* = 1, *s* = 1, n = 7) and birds (*v* = 1, *s* = 1, n = 1) (Fig 2). Non-native animals (including fish and crustaceans), introduced for commercial aquaculture (e.g. tilapia, carp, channel catfish) or imported as processed food (e.g. cod, conger, hake), also had relatively high species richness (*v* = 22, *s* = 16) and abundance (n = 132).

The most frequently reported species was the demersal and estuarine-dependent *tainha* (mullet, *Mugil liza*, Mugillidae, 11.3%, n = 187). Tainha appeared in the initial period of 1880–1889 and reached maximum relative frequency in 2010–2019. Tainha was followed by *camarão* (shrimp, Penaeidae, 9.3%, n = 154) and by pelagic and estuarine-demersal taxa including *anchova* (bluefish-enchova, *Pomatomus saltatrix*, Pomatomidae, 3.8%, n = 63), *baleia* (whale, several Mysticeti, 3.8%, n = 63), *cação/tubarão* (sharks, several Elasmobranchii,

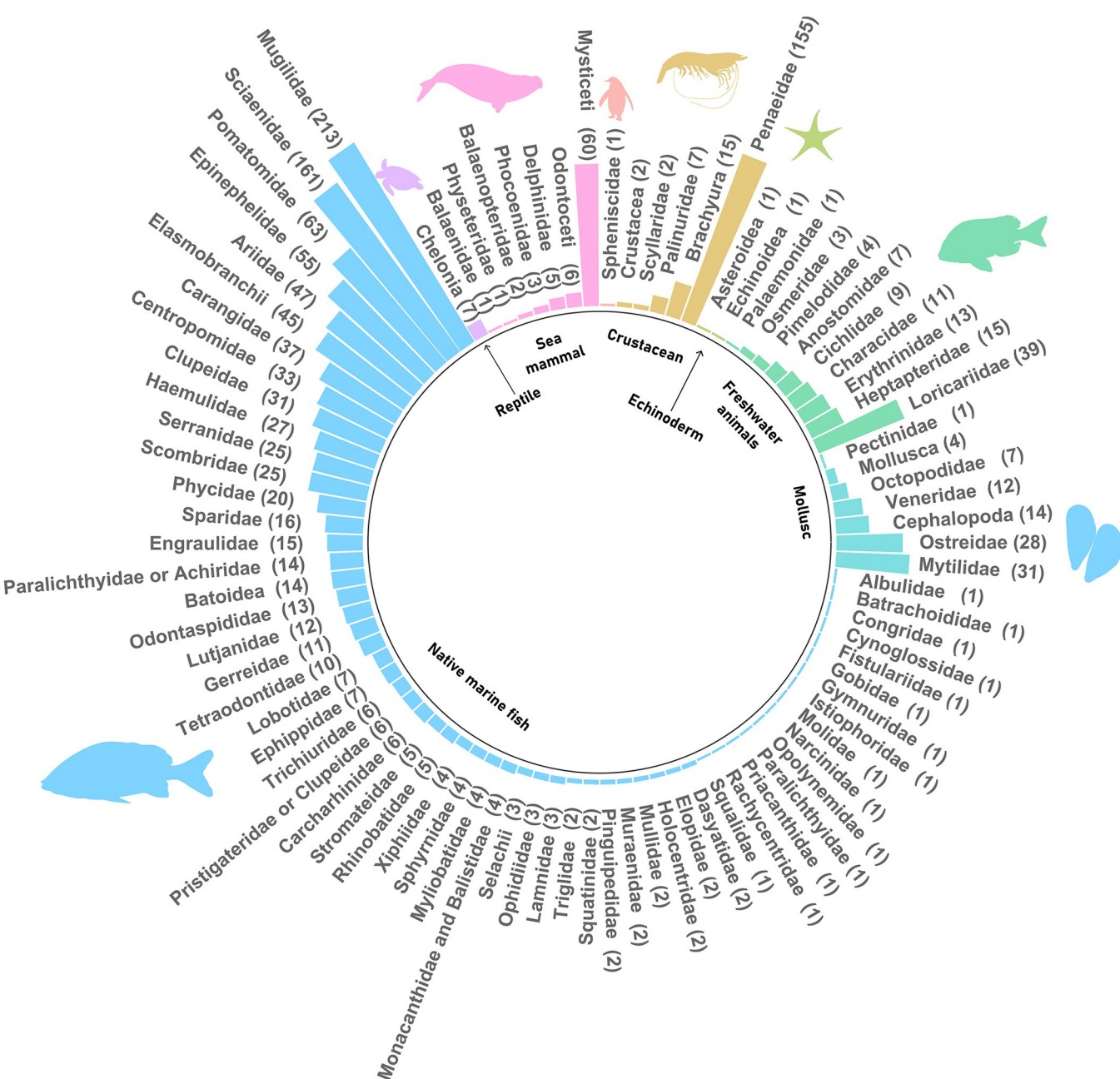

**Fig 2. Abundance of species (by family) reported in historical newspapers for locally exploited taxonomic groups between 1840 and 2019.**
Figure generated using R (https://www.r-project.org/), Inkscape (https://inkscape.org/) and GNU Image Manipulation Program (https://www.gimp.org/), on data publicly available from PhyloPic (http://phylopic.org/).

3.2%, n = 53), *bagre* (catfish, Ariidae, 2,8%, n = 47), *corvina* (whitemouth croaker, *Micropogonias furnieri*, Sciaenidae, 2.8%, n = 47), *cascudo* (armored catfish, Loricariidae, 2.2%, n = 37), *tilapia* (tilapia, *Oreochromis niloticus*, 2.1%, n = 35), *pescada* (drums/croakers, Sciaenidae, 1.9%, n = 31), *robalo* (fat snook, *Centropomus parallelus*, Centropomidae, 1.9%, n = 31), *ostra* (oyster, *Crassostrea rhizophorae*, Ostreidae, 1.7%, n = 28), *sardinha* (sardine, Clupeidae, 1.6%, n = 27), *garoupa* (dusky gruper, *Epinephelus marginatus*, Serranidae, 1.6%, n = 26), among others (Fig 2A). Some generic taxa were reported with distinct vernacular names, such as

*cação/tubarão* (e.g. *mangona, anequim, tintureira, baleeiro, cação gralha branca, c. salteador, c. azul*) and *camarão* (e.g. *camarão branco/legítimo/do corso, c. perereca, c. sete barbas, c. rosa/pistola, c. cabeça azul)*. When combined, the relative abundance of sharks and shrimps increased to 4.5% and 9.3%, respectively.

The number of items reporting marine animals using the keywords "*pesca*" (fishing) (n = 269) and "*peixe*" (fish) (n = 329) were significantly positively correlated with the number of newspapers for Santa Catarina available in the HDB archives (pesca r = 0.61; $R^2$ = 0.37; $p < 0.005$; peixe r = 0.52; $R^2$ = 0.27; $p < 0.02$), a pattern also observed by Gallardo et al. [43] (Fig 3A and 3B). Similarly, species richness and their abundances were also significantly positively correlated with the number of newspapers (species richness, r = 0.73; $R^2$ = 0.53; $p < 0.001$, frequency, r = 0.68; $R^2$ = 0.46; $p < 0.001$) in Santa Catarina (Fig 3C and 3D), indicating that, in absolute terms, our data is affected by the amount of digitised newspapers available in the HDB database.

When considering the relative proportion of distinct taxonomic categories and species, the results point to variations possibly expressing changes in socio-cultural, economic and market importance of aquatic animals through time (Fig 3E and 3F). Native marine fish and sea mammals had higher relative abundance and species richness between the decades of 1900–1909 to 1930–1939, followed by a decreasing trend toward the most recent decades (2000–2009 and 2010–2019). By contrast, the abundance and richness of invertebrates (molluscs, crustaceans) increased in the decades between 1900–1909 and 1930–1939, and again in 2000–2009 and 2010–2019. Freshwater fish and non-native species (both freshwater and marine), instead, showed the opposite trend, with higher relative abundances and species richness in the decades of 2000–2009 and 2010–2019 compared to previous decades.

The WATL shows the prevalence of high trophic level species over the last 180 years, including medium to large top predators such as *cherne* (groupers—*cherne-pintado* (snowy grouper), *garoupa* (dusky grouper), *mero* (goliath grouper)) and *tubarão/cação* (sharks—*cação mangona* (sand tiger shark), *c. baleeiro* (copper shark), *c. gralha branca* (whitetip shark)), among others (Fig 3G). However, a significant decline in WATL was observed between the late 19th and early 21st centuries (r = -0.56; $R^2$ = 0.31; $p < 0.05$). The ratio between piscivores and planktivores (Pis/Pla) was quite variable from 1880–1889 to 1960–1969 (ranging from 0.7 to 5), but maintained consistently low values (0.6 to 1) from 1980–1989 onwards (Fig 3H). These low trophic level species are mostly represented by *carapeba* (possibly *caratinga* (*mojarras*, *Eucinostomus* sp.), *gordinho* (American harvestfish, *Prepilus paru*), *parati* (white mullet, *Mugil curema*), and several Scianidae (*cangulo, pescadinhas, cangoá, guete*). Fluctuations in Pis/Pla did not correlate with time (r = -0.01; $R^2$ = 0.0002; $p > 0.05$), nevertheless, the results from combining these indicators point to a greater relative importance of high trophic level species (notably piscivores) from the late 19th to the mid 20th century, and a later decrease from the 1970's and 1980's. The high diversity and the relative frequency of *camarão* (shrimp), *ostra* (oyster), *marisco* (mussels), *siri* (crab), and *lula* (squid) since the late 19th century indicates that historical fisheries targeted organisms from all trophic levels. Interestingly, a significant increase of short-lived invertebrates (r = 0.58; $R^2$ = 0.35; $p < 0.05$) is observed from 1970–1979 onwards (Fig 3I), which is partially consistent with their increased captures by both artisanal and industrial fleets in the 1970's and 1980's in Santa Catarina (Fig 3J).

The reported organisms were associated with a range of fishing gear (n = 43), some of which only appeared in the late 19th and early 20th centuries (e.g. *chae-chae, caçoal*) (SI1). The diversity and abundance of fishing gear did not change as a function of time (*n* gear, r = 0.157; $R^2$ = 0.024; $p < 0.532$; frequency, r = 0.078; $R^2$ = 0.006; $p < 0.757$), but were significantly correlated with the number of newspapers (*n* gear, r = 0.641; $R^2$ = 0.411; $p < 0.005$; frequency, r = 0.806; $R^2$ = 0.649; $p < 0.001$). The broad category "*rede*" (net) dominated both in frequency

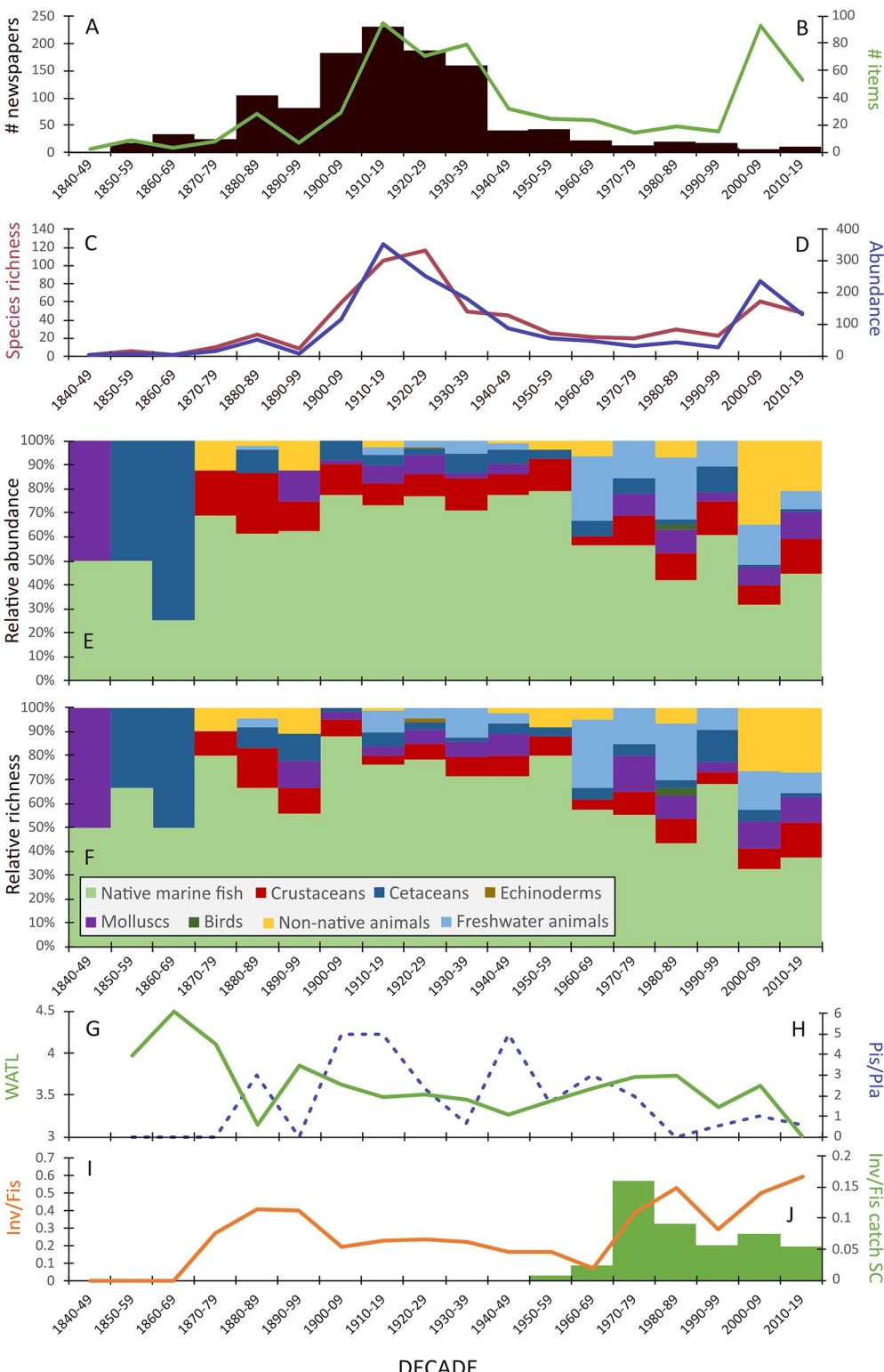

**Fig 3.** A) Decadal-scale trends in the number of newspapers for Santa Catarina available in the HDB archives (black bars) and B) news items (e.g. articles, opinion letters, regulations and sanctions) reporting aquatic animals using keywords fish and fishing (green line). C) Absolute abundance of species reported (blue line) and D) species richness as vernacular name (red line). E) Relative abundance of species and F) relative species richness according to main taxonomic groups. G) WATL of native marine fish, H) fish-based piscivores/planktivores (Pis/Pla), and I)

invertebrates/fish (Inv/Fis) from newspapers and J) from industrial and artisanal catch landing data (t) for Santa Catarina [20].

(n = 132) and diversity of gear (n = 18), including variations such as anchored, surface, bottom, drift, small mesh, large mesh, and trammel. According to ICMBio (Fishing Gear Type), these could be tentatively assigned to drift gillnets (*rede de emalhe*), which are some of most common fishing gear used to catch demersal species in south Brazil [65], and with documented use since the 1870's (*A Gazeta de Joinville*, 11 february 1879). Other relatively frequently used gear included "*aparelhos com anzol*" (use of a hook), "*tarrafa*" (cast nets), and "*arrasto*" (trawl, beach seine), to name a few.

## Discussion

### Species composition and the impact of early fisheries subsidies

Our study revealed that historical newspapers retain insightful information on the perceived value, diversity and abundance of marine animals in southern Brazil over the last 180 years. We identified nearly 300 aquatic animals, the majority of which were captured and traded by coastal communities and commercial fisheries in Santa Catarina since the 1840's, and others that were imported for direct consumption or aquaculture throughout the 20th and the early 21st centuries. The compiled information (e.g. frequency and richness), however, was unevenly distributed across the studied time interval, with fluctuations positively correlated with the number of digitised newspapers in the HDB database. This highlights the challenges of extracting absolute values to reconstruct ecological baselines from historical sources [50]. Therefore, in order to minimise sampling biases, we focus our discussion on the decades with greater amounts of information in the newspapers.

Significantly, 51% of our data (n = 307) was compiled from newspapers published between the decades of 1880–1889 and 1930–1939, which was a time interval of unprecedented political efforts to boost the commercial fishing sector in Brazil [66,67]. For example, in 1856, the Decree 876 of 10 September authorised the incorporation of companies for catching, salting and drying fish along the coast and rivers of the Brazilian Empire. Later in 1912, Decree 9672 of 17 July created the Fisheries Agency (*Inspetoria da Pesca*), replaced by the Biological Marine Station in 1915 (Estação de Biologia Marinha, Decree 11507 of 4 March), which promoted institutional actions for "studying and disseminating the natural resources of Brazilian waters, to develop them as much as possible and regulate their use". In 1933, Decree 23348 of 14 November established fish landing, processing and commercialization sites (*entreposto de pesca*), in order to increase efficiency of commercial fishing [66,68]. Surprisingly, little is known about the targeted species and the ecological implications of these public policies.

From the late 19th century, several fishing concessions were released to private investors in Santa Catarina (*A Regeneração*, 12 August 1884, 12 October 1888), while in the early 20th century, landing and seafood markets (*Entrepostos de Pesca*) were implemented through public investments in several cities along the Brazilian coast and increased state control of economic activities. Urban population growth increased demand for fish [69], enhancing market opportunities for private investors that greatly benefited from fiscal incentives in key urban centres such as Rio de Janeiro and Santos [70]. The extent of commercial fisheries development in Santa Catarina since the early 19th century is alluded to by export records from the city of Laguna for the years 1819 and 1820, which documented, among other fish species, the sale of more than 1 million catfish in 2 years to several Brazilian states (Bahia, Pernambuco, Rio de Janeiro) (*Revista Catarinense*, 1911). However, consolidated markets and long-trade networks

are better documented for later periods. Between 1910 and 1939, several locally caught fish (e.g. mullet, bluefish-enchova, shark, catfish, black and other types of drums) and invertebrates (e.g. shrimps, oysters, mussels) were reported in newspapers in relation to markets (22%, n = 133), landing reports (18%, n = 110), industrial fishing (14%, n = 83) and local food security (13%, n = 81). They were captured for direct human consumption (seafood) as well as for other byproducts including fishmeal, fish skin (as "sandpaper"), fish stomach and gut (as fertiliser) and oils, which were regularly traded in local and regional markets (*O Despertador*, 23 October 1880; *Republica*, 3 July 1903; *O Estado*, 22 December 1933). These local catches supplied the growing demands in urban centres (*A Gazeta*, 2 October 1935, but see also [71]), as well as the nutritional needs of small coastal communities (*O Estado*, 22 July 1915; *O Dia*, 3 August 1916; *A Noticia*, 18 September 1936; Republica, 22 October 1937). We suspect that commercial fisheries equally benefited from the specialised labour force, infrastructure and technology released with the decline of whaling in the late 19th century.

In the case of mullet, the most cited species with various distinct vernacular names ("*tainha*", "*tainhota*, "*tanhota*", "*tainha do corso*", "*tainha facão*", "*tainha-assu*"), intensive exploitations were attested since the late 19th century, when the species was primarily captured during its reproductive migration in winter (*A Regeneração*, 27 May and 7 June 1885; *O Estado*, 3 June 1915). Its fishing and processing (*escalada*, involving salting and desiccating) mobilised entire communities (*A Regeneração*,18 June 1885; *O Despertador*, 14 May 1881), and the catches were sold both locally (*O Estado*, 2 February and 1 March 1917) and in markets located hundreds of kilometres away, such as São Paulo (*O Estado*, 12 June 1923). Single catch events could consist of 30–60 thousand mullets (*O Estado*, 4 June 1941), even up to 100 thousand specimens (*O Estado*, 16 June 1933), which were captured using relatively simple fishing technology (gillnets and beach seine made of plant fibres). According to the captain Vieira da Rosa, on request of the Ministry of Agriculture in 1916, "*The most abundant fish in the state is tainha do corso* [winter mullet schools]. . .*It can be estimated that more than one million mullets are caught every winter. . .*" (*O Dia*, 8 August 1916). Considering the average weight of modern mullet during their winter reproductive migration (2 kg [31], but see also *O Estado*, 12 June 1923), this would correspond to 2000 t catch/winter, which is higher or comparable with artisanal landings/winter (May-July) in 2017 (1423 t), 2018 (1122 t) and 2019 (2522 t) for mullets in Santa Catarina state (data from PMAP-SC, http://pmap-sc.acad.univali.br/index.html). Although the productivity of winter migratory mullet schools can be affected by weather conditions [31,72], the reduced or equivalent landings in relation to the significantly increased fishing efforts of the more recent years supports the growing evidence that this species is currently overexploited in Santa Catarina [73].

Other species nowadays considered overexploited, such as *miraguaia* (black drummer, *Pogonias courbina*) and *corvina* (whitemouth croaker, *Micropogonias furnieri*) [24,25,74], were perceived as abundant and thus intensively targeted by coastal fisheries (*Sul Americano*, 12 August 1900, *A Noticia*, 18 February 1939, *O Estado*, 16 August 1930). High trophic level marine fish such as *anchova* (bluefish-enchova) and *cação-mangona* (sand tiger shark) were the main source of livelihood and financial capital to some coastal groups, as attested to in 1901: "*. . .the most important trade that Ribeirão makes at the head of the municipality, in the city of Florianópolis, is that of fish, which annually yields a hundred thousand reis, according to a general estimate. Two species of fish mainly occupy the activity of this honourable people, anchova* [bluefish-enchova] *and a variety of cação* [sharks], *known locally by the name of mangona* [sand tiger shark] *or xarque by the populations of the southernmost part of the island*" (*Sul Americano*, 22 July 1902).

Newspaper evidence of fishing intensification in the early 20th century is largely corroborated by the official reports of the Ministry of Agriculture, which enable linking the study

region discussed here (Santa Catarina) with the widespread ecological effects of historical fishing subsidies. In 1945, reports documented an increase of national fish production (fish byproduct) of 431% from 1934 (2112 t) to 1941 (11208 t). Brazil also shifted from importing 500 t of canned fish to exporting around 800 t of the product in 1943, mostly to South Africa and Guyana [75]. This suggests that commercial fisheries in Brazil benefited from reduced competition with foreign producers due to significant disruptions in fishing trade from the North Atlantic, North Sea and the Baltic during World War II [76]. The impact of these early subsides was also evident in the first official landing reports from the *Entreposto Federal de Pesca* in Rio de Janeiro, which indicated a 43% increase in landings (fish, invertebrates) from 1934 (13879 t) to 1940 (19913 t) [77]. As observed in most recent statistics [36,78], fisheries subsidies also caused a noticeable increase in overfishing and bycatching (prevalently juvenile specimens and small fish) in the first half of the 20th century in Brazil [43]. According to the Ministry of Agriculture, the confiscation of marine animals due to being under the minimum established size (bycatching) and deterioration increased 707% between 1934 (10580 t) and 1940 (85445 t) in the same *Entreposto Federal de Pesca*. Strikingly, confiscated fish and shellfish due to bycatching only, performed a dramatic increment of ~6000% in just five years, from 945 t in 1934 to 57585 t in 1939. Overfishing and bycatching, however, were not limited to marine species as official reports expressed growing concerns about the declining catches of freshwater animals. For example, decreasing landings of pirarucu (*Arapaima gigas*) in Belém (north Brazil) between 1919 (1.8 t) and 1938 (1.45 t) was largely attributed to overfishing. Other species such as manatees (*Trichechus* sp.) and arrau turtle (*Podocnemis expansa*) also saw a decrease in abundance in the Amazon by the 1930's, again attributed to intensive exploitation [77].

## Long-term change in species composition

When considering native marine fish only, the WATL shows consistently high values between the decades of 1900–1909 (WATL = 3.6) and 1940–1949 (WATL = 3.4), suggesting that top predators were highly desirable market items for quite some time, even though several species were already under noticeable fishing pressure and declining in wholesale markets [43]. Comparison with values for 2010–2019 reveals a sharp decline with the lowest values of the time series (WATL = 3.0). Interestingly, other independent indicators expressed the importance of high trophic level species in early decades, and their decreasing popularity in the most recent news. For example, planktivorous fish (Pis/Pla) and short-lived invertebrates (Inv/Fis) increased in their relative frequency of occurrence in newspapers from 1970 onwards at the expense of piscivorous fish. Significantly, long-term catch reconstructions for the coast of Santa Catarina show a similar trend in Inv/Fis, with catches of invertebrates increasing in the 1970's and 1980's, although this is followed by a downtrend in more recent decades [20]. This possibly reflects fiscal incentives provided to the fisheries by SUDEPE from 1967 onwards, along with fishing agreements between Brazil, Uruguay and Argentina, and the increased investment in shrimp and mollusc aquaculture [36,79–81]. For example, between 1966 and 1972 the number of fishing vessels employed in the catch of pink shrimp (*Farfantepenaeus* sp.) in Rio Grande (Rio Grande do Sul) increased from 80 to 396 [79]. In Santa Catarina shrimps were intensively exploited between 1970 and 1990 to the extent that coastal stocks declined considerably (*A Ponta*, July 1993, [80]). Moreover, increased market opportunities and public investment for farming of molluscs (e.g. mussels, oysters) and crustaceans (e.g. prawn and shrimps) accelerated production from the second half of the 20th century, and notably from the late 1980s [81,82] onward, which may explain their increased popularity in public media. Overall, the newspapers appear to capture the effect of market changes due to historical overexploitation [36] and increased fishing efforts [83] in Santa Catarina over the last 180 years.

From 2000–2009 the newspapers documented a substantial increase in freshwater and non-native species. Reports of freshwater species (mostly non-native) were primarily associated with aquaculture and imports for direct human consumption (*Correio do Povo*, 23 February 2000; *O Município*, 15 August 2008; *O Município*, 13–14 April 2017), along with recreational fishing (*Correio do Povo*, 3 May 1969) and re-population of rivers (*Correio do Povo*, 9–15 August 1980; *Correio do Povo*, 7 December 2002). Such a trend conceivably reflects a combination of factors, including the reduced contribution of local marine stocks to domestic consumption [84], changes in population demography, and increased investments in aquaculture in Brazil, notably between the 1990 and 2020 [82,85]. Significantly, the growing global importance of aquaculture in recent years, known as the "blue revolution", currently supplies half of all seafood for human consumption worldwide [86,87].

## Implications for marine conservation and Ocean Literacy principles

Our study reinforces the value of historical newspapers in providing insightful information on long-term species composition, perceived abundance and the drivers of change in marine biodiversity in Brazil. However, contrary to other records such as official landing statistics, species composition in newspapers reflect distinct socio-ecological contexts, spanning from production to consumption systems, as well as considerations on the cultural and ecological values of some species, which are subject to evolving political, ideological and economic agendas, notably in social contexts with high levels of illiteracy as was the case in the late 19th and early 20th centuries in Brazil [52,88,89]. Moreover, we acknowledge that information from newspapers is often fragmented, with substantial temporal and spatial gaps. Nevertheless, their use in historical ecology has proved fruitful [42,43] and here we show their potential to uncover species composition (caught or observed) and related social, economic and market values for decades prior to monitoring programmes and scientific observations.

Specifically, the late 19th and early 20th centuries witnessed the increased commoditization of marine resources in south Brazil. Profit- and efficiency-seeking fisheries policies accelerated the "economy of things" at the expense of the "economy of relationships" between communities and environments [90], exacerbating unsustainable exploitation and weak governance in the following decades. The early 20th century in particular can be considered the "incubation chamber" of fisheries industrialization in the country, but knowledge gaps on species composition and the ecological impacts of these early subsides may lead to the misconception that the commercial sector was limited in extent and volume, and that most fisheries in the late 19th and early 20th centuries were for local subsistence or incipient local markets. Such views impinge upon our understanding of the extent to which marine and ocean systems have been altered by anthropogenic activities.

Newspapers provide valuable insights into the changing nature of long-term human interaction with marine biodiversity, helping to track the origin and evolution of Traditional Ecological Knowledge. This knowledge plays a fundamental role in fisheries management [91], biological conservation [92,93] and coastal livelihoods [68,70,94], and newspapers reveal that it developed in a historical context of conflicts with institutions and political and economic actors, who often associated local fishers with backwardness, poverty, and destitution, and attributed the richness of their fishing practices with detrimental impacts and economic underdevelopment [43]. The establishment of the Fishery Inspection Agency in 1912 is an example of how political institutions attempted to override local fisheries knowledge with "modern" knowledge in order to increase efficiency and unlock the market potential of aquatic resources in the country [95]: "*Until now, unfortunately, this industry* [fishing], *among us, has not passed from the backward and old primitive processes, the hook, the trawler and, lately, the*

*dynamite, ruthlessly damaging the creations of the coasts and almost extinguishing several lacustrine, fluvial and maritime species. The improved fishing gear that is currently being used, with great results, without impoverishing the waters in which they are exercised, and mechanical fishing by means of steamboats have not yet been used. . .".* This market logic has for decades contributed to dismantling traditional forms of socio-ecological interactions [70]. Their legacy can still be perceived among modern small-scale fisheries [96] and the social and political context around them [97].

Given their role in shaping public opinion through time, historical newspapers offer a rich repository of shifting ecological baselines. Therefore they are valuable resources to inform the ambitious Mission of the UN Decade of Ocean Science for Sustainable Development to promote "Transformative ocean science solutions for sustainable development, connecting people and our ocean". A core driving concept advanced by the UN Ocean Decade is to push countries to initiate Ocean Literacy programs across the globe, in order to promote better public understanding, communication and informed policy decisions for ocean sustainability [98]. By resurrecting previously hidden social-ecological baselines, historical records can help Ocean Literacy programs advance on essential principles and fundamental concepts about the functioning of the ocean [16], including: clarity on the diversity of life and ecosystems oceans once supported (principle 5); acknowledgment of deep, historical interconnections humans have held with the ocean (principle 6); and more realistic baselines about the level of human exploration of coastal marine environments (principle 7).

## Conclusion

Humans have depended on marine ecosystems as a source of food and livelihood for thousands of years along the Brazilian coast [99], a process that has laid down the foundation of a diversity of fishing cultures in the region [68,70]. Over the last few decades, increased fishing demand, overoptimistic cycles of profit-driven subsidy programmes and weak governance models intensified commercial exploitations, leading to unprecedented levels of catches and the decline of a range of stocks [24,25,36]. This work advances our understanding of this convoluted historical process by expanding the current knowledge of captured and consumed marine animals in southern Brazil by nearly two centuries, covering decades predating national official landing reports and market information. Our results highlight the utility of newspaper records in resurrecting previously lost marine social-ecological baselines, with a high potential to support Ocean Literacy programs to rightfully contextualise historical realities when shaping sustainability pathways with ocean stakeholders at all levels.

## Supporting information

**S1 File. Historical newspapers sourced from the Brazilian Digital Newspapers and Periodicals Library (*Hemeroteca Digital Brasileira, Fundação Biblioteca Nacional, BNDigital),* including reported marine animals and fishing gear.**
(XLSX)

## Acknowledgments

We would like to thank Leopoldo Cavaleri Gerhardinger and Krista McGrath for their insightful comments on the first draft of the manuscript. The authors are also grateful to Natalia Hanazaki and the Laboratory of Human Ecology and Ethnobotany (Universidade Federal de Santa Catarina, Brazil).

## Author Contributions

**Conceptualization:** Dannieli Firme Herbst, Luiz Geraldo Silva, André Carlo Colonese.

**Data curation:** Dannieli Firme Herbst, Thiago Fossile, André Carlo Colonese.

**Formal analysis:** Dannieli Firme Herbst, Jara Rampon, Bruna Baleeiro, Thiago Fossile, André Carlo Colonese.

**Funding acquisition:** André Carlo Colonese.

**Investigation:** Dannieli Firme Herbst, Jara Rampon, Bruna Baleeiro, André Carlo Colonese.

**Methodology:** Luiz Geraldo Silva.

**Project administration:** André Carlo Colonese.

**Resources:** André Carlo Colonese.

**Supervision:** André Carlo Colonese.

**Writing – original draft:** Dannieli Firme Herbst, André Carlo Colonese.

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
