## [Decision Letter · Decision Letter 0]

18 Jan 2023

PONE-D-22-32178180 years of marine animal diversity as perceived by public media in southern BrazilPLOS ONE

Dear Dr. Herbst,

Thank you for submitting your manuscript to PLOS ONE. After careful consideration, we feel that it has merit but does not fully meet PLOS ONE’s publication criteria as it currently stands. Therefore, we invite you to submit a revised version of the manuscript that addresses the points raised during the review process.

Please consider the comments of reviewer 1 regarding potential use of official fishery statistics to support and complement your findings.  This should be possible for some of the more recent decades though probably not for the earliest decades where official records may be more limited or non-existent.  Also please make changes according to comments of both reviewers regarding several points for clarification and readability.

We look forward to receiving your revised manuscript.

Kind regards,

Brian R. MacKenzie, Ph. D.

Academic Editor

PLOS ONE

Journal Requirements:

3. We note that Figure 1 in your submission contain map image which may be copyrighted. All PLOS content is published under the Creative Commons Attribution License (CC BY 4.0), which means that the manuscript, images, and Supporting Information files will be freely available online, and any third party is permitted to access, download, copy, distribute, and use these materials in any way, even commercially, with proper attribution. For these reasons, we cannot publish previously copyrighted maps or satellite images created using proprietary data, such as Google software (Google Maps, Street View, and Earth). For more information, see our copyright guidelines: http://journals.plos.org/plosone/s/licenses-and-copyright.

4. We note that Figure 2 in your submission contain copyrighted images. All PLOS content is published under the Creative Commons Attribution License (CC BY 4.0), which means that the manuscript, images, and Supporting Information files will be freely available online, and any third party is permitted to access, download, copy, distribute, and use these materials in any way, even commercially, with proper attribution. For more information, see our copyright guidelines: http://journals.plos.org/plosone/s/licenses-and-copyright.

Reviewers' comments:

Reviewer's Responses to Questions

**Comments to the Author**

1. Is the manuscript technically sound, and do the data support the conclusions?

Reviewer #1: Partly

Reviewer #2: Yes

2. Has the statistical analysis been performed appropriately and rigorously? 

Reviewer #1: Yes

Reviewer #2: Yes

3. Have the authors made all data underlying the findings in their manuscript fully available?

Reviewer #1: Yes

Reviewer #2: Yes

4. Is the manuscript presented in an intelligible fashion and written in standard English?

Reviewer #1: Yes

Reviewer #2: Yes

5. Review Comments to the Author

Reviewer #1: Overview:

The work brings a historical rescue of fishing data for one of the largest production regions at national level (Santa Catarina). Although the approach does not bring methodological novelties, the work is relevant considering the scarcity/fragmentation of information at the local/national level.

The work is well structured, however, below are some suggestions to mitigate the bias related to the search for information in newspapers. I believe that authors should look for ways to compare these historical records with official fishing statistics existing at the regional level. In this way, it will be possible to assess whether, in this specific study, newspaper records reflect (or not) official statistics, at least for the most recent periods (20th and 21st centuries).

Methods:

Page 5, Lines 131-132: It is necessary to indicate which were the keywords used for the search of all the resources (fish, crustaceans, molluscs, mammals, reptiles), since the results show several resources and not just 'fish'.

P 6, L 141-143: Please include how the search was performed on this platform (keywords, etc.).

P 6, L 165-167: Did you use the ratio between abundance, catch weight (landings)? Make it more clear to the reader.

Results:

P 11, L 272-273: This result may be related to the projects that promoted the implementation of the industrial productive segment of shrimp in Santa Catarina in the 1970s (e.g., SUDEPE). In addition to projects aimed at growing mangrove oysters in association with shrimp in the 1980s. In this way, there is a need for a greater deepening in the discussion about public policies directed to the productive sector that promoted the cultivation of these aquatic organisms, bearing in mind that this may influence the publicity of journalistic articles on the subject.

Discussion:

P 15, L 397-421: This paragraph is outside the scope of the study, as it provides an overview of fishing at the national level, while the study focuses on the regional level (Santa Catarina). Is the pattern described at the national level also observed at the regional level? I suggest that this paragraph be revised according to the purpose of the study.

P 16, L 433-435: this may be related to what was pointed out above (see comment in Results section).

P 17, L 451-452: Is this trend also observed in Santa Catarina?

P 18, L 505-506: As the authors themselves recognize above, the records obtained from newspapers can suffer from several biases. One possibility of mitigating this bias and verifying whether newspaper records actually reflect catches is through cross-analysis with historical sources of fisheries statistics. I agree that statistical records are incomplete, fragmented and scattered, but it is possible to use some official data sources that can serve to assess general trends. For example, it is possible to use ICMBio/CEPSUL records (https://www.icmbio.gov.br/cepsul/biblioteca/acervo-digital/38-download/artigos-cientificos/112-artigos-cientificos.html), IBGE (https://biblioteca.ibge.gov.br/biblioteca-catalogo.html?id=7132&view=detalhes) and IBAMA (https://www.gov.br/ibama/pt-br/assuntos/biodiversidade/biodiversidade-aquatica/gestao-da-biodiversidade-aquatica/estatistica-pesqueira). I believe it might be possible to check the general trend between invertebrates/fishes with this database. Thus, we can assess whether the methodology used in the work reflects (at least partially) the official statistics (albeit fragmented).

Figure 3A,B: include in the legend the meaning of the colors present in the figure (A, B). There are two axes but three colors.

Reviewer #2: This is an interesting paper that is scientifically sound. My comments are minor and mostly relate to clarifying parts of the text to ensure accurate interpretation of methods and results by readers.

L43: ‘…while ensuring fundamental ecosystem functions, structures and services’ Please provide a reference for this.

L50: ‘This sort of historical amnesia creates misconceptions…’ This sentence is very long. Suggest splitting up to improve readability.

L107: ‘…scrutinised more than twenty thousand digitised editions in the Brazilian Digital Newspapers and Periodicals Library (Hemeroteca Digital Brasileira)’. What do you mean by ‘scrutinised’? If you did not examine at all of these (rather, you conducted key-word searches via OCR across this number of newspapers) this phrasing is rather misinformative.

Fig 1 Legend: The coloured dots should be labelled as >1 – 10 reports, >10 – 20 reports… >30 – 200 reports etc for greater clarity.

Also, ‘The colour gradient of the localities represents the number of reports IDENTIFIED per municipality’.

L133: what dates did you undertake data collection? You mention dates in the results section but it is not clear if these were dates of collection – this is important to know given that more newspaper articles/editions are being uploaded to digital collections all the time.

Given you identified animals from articles initially located by keywords fishing and fish, how likely is it that you’ve missed accounts of species that were gathered or cultured, or otherwise exploited/identified using alternative words to ‘fishing’ or ‘fish’?

L159: Suggest ‘Species richness REPORTED by decade’ for clarity.

L184: Suggest restructuring of this paragraph as it initially reads that you individually screened the >20,000 articles, rather than identify the ca. 1200 from SC which were then individually screened. Regarding the 4.7% of the total newspapers you mention, were the 95.3% of papers available from the SC region but that did not get identified from the OCR searches (i.e. did not mention fishing or fish)? In general, this paragraph is unclear.

L198: I.e. ‘(v = 203, s = 107)’: Do you mean that a total of 310 species were identified, or are these two numbers not additive? Same question for the other numbers that follow.

L199: ‘frequency (n = 1,041)’: Do you mean frequency of mentions?

L234: ‘The number of items reporting marine animals using the keywords fishing (n = 269) and fish (n = 329) were significantly positively correlated with the number of newspapers (fishing r = 0.61; R2 = 0.37; p < 0.005; fish r = 0.52; R2 = 0.27; p < 0.02)’ Can you explain this more clearly? Are you referring to the total number of newspapers available (i.e. published online) in the region, or those identified through OCR searches as containing the words ‘fishing’ or ‘fish’? This is clarified somewhat by the legend in Fig 3 but it needs to be clear in the text, too.

Fig. 3A and 3B – there are three colours in this panel figure, the black I assume correlates to the black text on the y axis descriptor (fig 3a), but there are also two red block colours in this figure and I am now sure how these align with B and the red text on the secondary y axis (#items). If these are aggregated per the legend description, why two red colours? Also, what does ‘items’ mean? Is this referring to individual articles within a newspaper edition?

Fig. 3C-E – there seems to be very little difference between absolute species counts and species richness. It may be worth briefly repeating in the text (or legend description) the difference between these two measures, as it gets a bit confusing when examining the figure.

In general, the way your results are written it reads as absolute species richness or species numbers, when actually it is richness or numbers reported in newspapers. I think this needs to be stated more clearly throughout the results. I.e., L256: By contrast, the richness and frequency of invertebrates REPORTED increased… L263: The WATL shows the prevalence of high trophic level species REPORTED over the last 180 years… L287: The diversity and frequency of fishing gear REPORTED did not change…

L466: you state the potential for newspapers to uncover species composition, better to clarify here that it is composition of fishery catch and/or landings? This of course reflects to some extent underlying ecological composition, but acknowledges the influences of social (in terms of what is reported) and market forces and technological trends (in terms of what is caught) on newspaper reporting.

L473: misspelling of ‘unsustainable’.

L482: The relevance of this paragraph on LEK to the wider paper is unclear and it doesn’t tie in overly well. The historical context of conflicts you mention regarding LEK will undoubtedly also play out in popular media, it being a window into past cultures. But I’m not sure if you’re mentioning this to say that this is indeed the case with newspapers or to say that historical newspaper sources can mitigate this issue?

L485. Your definition of LEK seems more akin to traditional ecological knowledge in terms of intergenerational transmission. While I am aware that TEK can be defined as a form of LEK, I think this needs to be clarified.

6. PLOS authors have the option to publish the peer review history of their article (what does this mean?). If published, this will include your full peer review and any attached files.

Reviewer #1: **Yes: **Cleverson Zapelini

Reviewer #2: No

---

## [Author Response · Author response to Decision Letter 0]

26 Feb 2023

PONE-D-22-32178

180 years of marine animal diversity as perceived by public media in southern Brazil

PLOS ONE

Dear Academic Editor

Thank you for considering our manuscript entitled 180 years of marine animal diversity as perceived by public media in southern Brazil, by Herbst et al. for publication in PLOS ONE.

We have now completed the revision of our manuscript according to the comments made by Reviewer #1 and #2 and the Academic Editor. We would like to thank you in the quality of Academic Editor and both reviewers for your constructive comments which helped to improve the quality of the manuscript and to clarify some of our statements.

Please find below our rebuttal. All changes to the manuscript are marked in red to facilitate their identification.

We would like to update our financial statement with the statement below:

This work was funded by the ERC Consolidator project TRADITION, which has received funding from the European Research Council (ERC) under the European Union’s Horizon 2020 research and innovation programme under Grant Agreement No 817911. This work was also funded by EarlyFoods (Evolution and impact of early food production systems), 2021 SGR 00527. This work contributes to the ICTA-UAB “María de Maeztu'' Programme for Units of Excellence of the Spanish Ministry of Science and Innovation (CEX2019-000940-M). 

We hope the manuscript can now be accepted for publication.

Kind regards,

Herbst et al.

Dear Dr. Herbst,

Thank you for submitting your manuscript to PLOS ONE. After careful consideration, we feel that it has merit but does not fully meet PLOS ONE’s publication criteria as it currently stands. Therefore, we invite you to submit a revised version of the manuscript that addresses the points raised during the review process.

Please consider the comments of reviewer 1 regarding potential use of official fishery statistics to support and complement your findings. This should be possible for some of the more recent decades though probably not for the earliest decades where official records may be more limited or non-existent. Also please make changes according to comments of both reviewers regarding several points for clarification and readability.

● A rebuttal letter that responds to each point raised by the academic editor and reviewer(s). You should upload this letter as a separate file labeled 'Response to Reviewers'.

● A marked-up copy of your manuscript that highlights changes made to the original version. You should upload this as a separate file labeled 'Revised Manuscript with Track Changes'.

● An unmarked version of your revised paper without tracked changes. You should upload this as a separate file labeled 'Manuscript'.

We look forward to receiving your revised manuscript.

Kind regards,

Brian R. MacKenzie, Ph. D.

Academic Editor

PLOS ONE

Journal Requirements:

3. We note that Figure 1 in your submission contain map image which may be copyrighted. All PLOS content is published under the Creative Commons Attribution License (CC BY 4.0), which means that the manuscript, images, and Supporting Information files will be freely available online, and any third party is permitted to access, download, copy, distribute, and use these materials in any way, even commercially, with proper attribution. For these reasons, we cannot publish previously copyrighted maps or satellite images created using proprietary data, such as Google software (Google Maps, Street View, and Earth). For more information, see our copyright guidelines: http://journals.plos.org/plosone/s/licenses-and-copyright.

R: The figures were generated using free access images (Public Domain or CC BY 4.0 licence) and references to their sources are included in the revised figure captions. We provide more information about the individual licences below:

● Natural Earth (https://www.naturalearthdata.com/about/terms-of-use/)

● National Institute for Space Research - INPE (http://terrabrasilis.dpi.inpe.br/citacoes-e-licenca-de-uso/)

● CGIAR Consortium for Spatial Information (https://cgiarcsi.community/data/srtm-90m-digital-elevation-database-v4-1/)

4. We note that Figure 2 in your submission contain copyrighted images. All PLOS content is published under the Creative Commons Attribution License (CC BY 4.0), which means that the manuscript, images, and Supporting Information files will be freely available online, and any third party is permitted to access, download, copy, distribute, and use these materials in any way, even commercially, with proper attribution. For more information, see our copyright guidelines: http://journals.plos.org/plosone/s/licenses-and-copyright.

R: The figures were generated using free access images (Public Domain or CC BY 4.0 licence) and references to their sources are included in the revised figure captions. We provide more information about the individual licences below:

● PhyloPic (http://phylopic.org/)

R: Done, captions for Supporting Information files now reported

Reviewers' comments:

Reviewer's Responses to Questions

Comments to the Author

1. Is the manuscript technically sound, and do the data support the conclusions?

Reviewer #1: Partly

Reviewer #2: Yes

2. Has the statistical analysis been performed appropriately and rigorously?

Reviewer #1: Yes

Reviewer #2: Yes

3. Have the authors made all data underlying the findings in their manuscript fully available?

Reviewer #1: Yes

Reviewer #2: Yes

4. Is the manuscript presented in an intelligible fashion and written in standard English?

Reviewer #1: Yes

Reviewer #2: Yes

5. Review Comments to the Author

Reviewer #1: Overview:

The work brings a historical rescue of fishing data for one of the largest production regions at national level (Santa Catarina). Although the approach does not bring methodological novelties, the work is relevant considering the scarcity/fragmentation of information at the local/national level.

The work is well structured, however, below are some suggestions to mitigate the bias related to the search for information in newspapers. I believe that authors should look for ways to compare these historical records with official fishing statistics existing at the regional level. In this way, it will be possible to assess whether, in this specific study, newspaper records reflect (or not) official statistics, at least for the most recent periods (20th and 21st centuries).

R: We thank the reviewer #1 for his/her/their suggestion. We want to highlight that the emphasis of our paper is on the perception of mass media on marine biodiversity changes, for this reason, we believe that attempts to validate data from newspapers (and derivatives) with independent landing records may risk to open more questions than answers, and the nature of these questions may remain largely contentious. Moreover, official fishing statistics could also carry other analytical and statistical biases complex to resolve, as exemplified by Freire et at [1,2]. However, following suggestions by the Reviewer #1 we have now compared our Inv/Fis index with the estimated index from reconstructed and officially reported industrial and artisanal catch data (volume in tons) for Santa Catarina from 1950 to 2015 [1] (see lines 184 to 186, page 6). We have revised the text which now reads (lines 474 to 488, page 15): “Significantly, long-term catch reconstructions for the coast of Santa Catarina show a similar trend in Inv/Fis, with catches of invertebrates increasing in the 1970’s and 1980’s, although this is followed by a downtrend in more recent decades [1]. This possibly reflects fiscal incentives provided to the fisheries by SUDEPE from 1967 onwards, along with fishing agreements between Brazil, Uruguay and Argentina, and the increased investment in shrimp and mollusc aquaculture [3–6]. For example, between 1966 and 1972 the number of fishing vessels employed in the catch of pink shrimp (Farfantepenaeus sp.) in Rio Grande (Rio Grande do Sul) increased from 80 to 396[4]. In Santa Catarina shrimps were intensively exploited between 1970 and 1990 to the extent that coastal stocks declined considerably (A Ponta, July 1993,[5]). Moreover, increased market opportunities and public investment for farming of molluscs (e.g. mussels, oysters) and crustaceans (e.g. prawn and shrimps) accelerated production from the second half of the 20th century, and notably from the late 1980s[6,7] onward, which may explain their increased popularity in public media.

Methods:

Page 5, Lines 131-132: It is necessary to indicate which were the keywords used for the search of all the resources (fish, crustaceans, molluscs, mammals, reptiles), since the results show several resources and not just 'fish'.

R: The keywords used are those presented in the original text (lines 131 and 132, page 5 of the original manuscript): “pesca” (fishing) and "peixe" (fish). We have added the following texts to clarify the importance of these two keywords:

Material and Methods (Lines 141 to 151, page 5)

“The keywords “pesca” (fishing) and "peixe" (fish) were chosen for the following reasons: i) The keyword “pesca” captured several types of marine resources, and also captured the word "pescado" which categorically addresses different types of resources exploited by fisheries; ii) within Brazilian folk taxonomy (ethnotaxonomy), various marine animals ("pescado") are classified as “fish”[8,9]; iii) the popular association of fish such as shrimp, lobster, oysters, octopus and squid with crustaceans and molluscs is something very recent and thus this nomenclature is infrequently used in everyday life; iv) given the large diversity of organisms captured by coastal fisheries in Brazil and the general scope of the paper (not targeting a specific species), it would be unfeasible to extend the keywords to all potential captured organisms.”

Results (Lines 215 to 217, page 7)

“The keywords “pesca” (fishing) and "peixe" (fish) captured animals from different taxonomic groups (fish, crustaceans, molluscs, mammals, reptiles), which is significant since in folk taxonomy several taxa are also considered as “fish”.”

P 6, L 141-143: Please include how the search was performed on this platform (keywords, etc.).

R: The method is reported in detail in Galardo et al. [10], and succinctly described in the Material and Methods. We have now expanded it to clarify the steps and process, please see lines 131 to 169 (Pages 5 and 6).

P 6, L 165-167: Did you use the ratio between abundance, catch weight (landings)? Make it more clear to the reader.

R: We used the absolute abundance of species per decade. The sentence has been revised for clarification. Now it's reads (lines 176 to 179, page 6):

“In addition, we computed fish-based piscivores/planktivores (Pis/Pla)[11] and invertebrates/fish (Inv/Fis) to explore changes in overall catch composition through time. For this we used the absolute abundance of species reported in newspapers per decade.”

Results:

P 11, L 272-273: This result may be related to the projects that promoted the implementation of the industrial productive segment of shrimp in Santa Catarina in the 1970s (e.g., SUDEPE). In addition to projects aimed at growing mangrove oysters in association with shrimp in the 1980s. In this way, there is a need for a greater deepening in the discussion about public policies directed to the productive sector that promoted the cultivation of these aquatic organisms, bearing in mind that this may influence the publicity of journalistic articles on the subject.

R: Yes, we agree and have now expanded the argument in the discussion, see please lines 477 to 488, page 15.

The impact of public policies, as well as of evolving ideological and economic agendas on the quality and number of news is acknowledged in lines 459 and 464 (page 17 original version). We have expanded the argument (lines 507 to 515, page 16):

“Our study reinforces the value of historical newspapers in providing insightful information on long-term species composition, perceived abundance and the drivers of change in marine biodiversity in Brazil. However, contrary to other records such as official landing statistics, species composition in newspapers reflect distinct socio-ecological contexts, spanning from production to consumption systems, as well as considerations on the cultural and ecological values of some species, which are subject to evolving political, ideological and economic agendas, notably in social contexts with high levels of illiteracy as was the case in the late 19th and early 20th centuries in Brazil[12–14].”

Discussion:

P 15, L 397-421: This paragraph is outside the scope of the study, as it provides an overview of fishing at the national level, while the study focuses on the regional level (Santa Catarina). Is the pattern described at the national level also observed at the regional level? I suggest that this paragraph be revised according to the purpose of the study.

R: We disagree with the Reviewer #1. The evidence discussed in this paragraph is highly relevant and within the scope of the paper. It enabled us to scale up our showcase (Santa Catarina) with the widespread ecological effects of increased fishing effort in Brazil. In the discussion, by considering the results of the reports from the Ministry of Agriculture (responsible for fisheries management in Brazil during the analysed timeframe), we enlarge the context of fishing to national and international levels, which is also crucial for tracking the development of fishing in Santa Catarina. We have slightly modified the text to emphasise its relevance and now it reads (lines 534 to 537, page 14):

“Newspaper evidence of fishing intensification in the early 20th century is largely corroborated by the official reports of the Ministry of Agriculture, which enable linking the study region discussed here (Santa Catarina) with the widespread ecological effects of historical fishing subsidies.” 

P 16, L 433-435: this may be related to what was pointed out above (see comment in Results section).

R: Thanks, the comment has been addressed above.

P 17, L 451-452: Is this trend also observed in Santa Catarina?

R: Yes, this references mentioned analysis and trend of Santa Catarina.

P 18, L 505-506: As the authors themselves recognize above, the records obtained from newspapers can suffer from several biases. One possibility of mitigating this bias and verifying whether newspaper records actually reflect catches is through cross-analysis with historical sources of fisheries statistics. I agree that statistical records are incomplete, fragmented and scattered, but it is possible to use some official data sources that can serve to assess general trends. For example, it is possible to use ICMBio/CEPSUL records (https://www.icmbio.gov.br/cepsul/biblioteca/acervo-digital/38-download/artigos-cientificos/112-artigos-cientificos.html), IBGE (https://biblioteca.ibge.gov.br/biblioteca-catalogo.html?id=7132&view=detalhes) and IBAMA (https://www.gov.br/ibama/pt-br/assuntos/biodiversidade/biodiversidade-aquatica/gestao-da-biodiversidade-aquatica/estatistica-pesqueira). I believe it might be possible to check the general trend between invertebrates/fishes with this database. Thus, we can assess whether the methodology used in the work reflects (at least partially) the official statistics (albeit fragmented).

R: Done, comparison of Inv/Fis was performed with data published in Freire et al. [1] that includes data of different sources. See previous comments 

Figure 3A,B: include in the legend the meaning of the colors present in the figure (A, B). There are two axes but three colors.

R: Done, colours made clear now.

Reviewer #2: This is an interesting paper that is scientifically sound. My comments are minor and mostly relate to clarifying parts of the text to ensure accurate interpretation of methods and results by readers.

L43: ‘…while ensuring fundamental ecosystem functions, structures and services’ Please provide a reference for this.

R: Done!

L50: ‘This sort of historical amnesia creates misconceptions…’ This sentence is very long. Suggest splitting up to improve readability.

R: Done, it now reads (lines 49 to 55 page 2): 

“This form of historical amnesia creates misconceptions that have far-reaching consequences for sustainability and conservation actions. For example, it may conceivably increase tolerance for progressive environmental degradation and contribute to setting inappropriate sustainability targets and responses by ocean stakeholders; and hence underpin their expectations as to what is a desirable and achievable state of social-ecological systems[15–17].”

L107: ‘…scrutinised more than twenty thousand digitised editions in the Brazilian Digital Newspapers and Periodicals Library (Hemeroteca Digital Brasileira)’. What do you mean by ‘scrutinised’? If you did not examine all of these (rather, you conducted key-word searches via OCR across this number of newspapers) this phrasing is rather misinformative.

R: The search via OCR with the keywords "pesca" and "fish" found 9,190 and 14,168 matches respectively (see in the results), in other words, more than twenty thousand digitised editions (newspaper reports) were accessed with these keywords. Scrutinised means that we examined closely and thoroughly all the news items.

Fig 1 Legend: The coloured dots should be labelled as >1 – 10 reports, >10 – 20 reports… >30 – 200 reports etc for greater clarity.

Also, ‘The colour gradient of the localities represents the number of reports IDENTIFIED per municipality’.

R: Thank you for your suggestion! Done!

L133: what dates did you undertake data collection? You mention dates in the results section but it is not clear if these were dates of collection – this is important to know given that more newspaper articles/editions are being uploaded to digital collections all the time.

R: From March 2021 to March 2022. This information has now been added to the text (Lines 196 to 197, page 6)

Given you identified animals from articles initially located by keywords fishing and fish, how likely is it that you’ve missed accounts of species that were gathered or cultured, or otherwise exploited/identified using alternative words to ‘fishing’ or ‘fish’?

R: It is possible that we could have had a more extensive database. However, within a long list of complementary and/or alternative keywords, the use of two generic keywords (fishing and fish) that can be applicable elsewhere covered a broad range of marine organisms. 

L159: Suggest ‘Species richness REPORTED by decade’ for clarity

R: Done!

L184: Suggest restructuring of this paragraph as it initially reads that you individually screened the >20,000 articles, rather than identify the ca. 1200 from SC which were then individually screened. Regarding the 4.7% of the total newspapers you mention, were the 95.3% of papers available from the SC region but that did not get identified from the OCR searches (i.e. did not mention fishing or fish)? In general, this paragraph is unclear.

R: The text has been revised for clarity. About the other 95.3% of newspapers available from SC did not meet the inclusion criteria, namely i) the items focused on coastal and ocean areas, including estuaries and coastal lagoons, or coastal rivers; and ii) the items reported marine animals (e.g. fish, crustaceans, molluscs, mammals, reptiles), including non-native taxa (Supplementary Information 1, SI1).

L198: I.e. ‘(v = 203, s = 107)’: Do you mean that a total of 310 species were identified, or are these two numbers not additive? Same question for the other numbers that follow.

R: These are two independent classifications, therefore they are not additive.

We have amended the text for clarification, and now it reads (lines 215 to 227, pages 7 and 8): 

“The keywords “pesca” (fishing) and "peixe" (fish) captured animals from different taxonomic groups (fish, crustaceans, molluscs, mammals, reptiles), which is significant since in folk taxonomy several taxa are also considered as “fish”. Native and non-native animals were mostly reported as vernacular (v) and, to a lesser extent, as scientific (s) names, the latter sporadically appearing in more recent decades. We found 1655 mentions of marine animals, totaling 277 and 203 vernacular and scientific taxa respectively (SI2). As reported by Feire and Pauly[18], we also found evidence that a single folk nomenclature was used for several species (e.g. Hypanus sp., Dasyatis sp., Fontitrygon sp., Bathytoshia sp. = raia prego), and several folk nomenclatures were also used for a single species (e.g. Mugil liza = tainha, tainhota, tanhota, do corso, facão, tainha-assú). As a consequence, attempts to correlate scientific and vernacular names were complex and not always possible with an equal level of confidence.”

L199: ‘frequency (n = 1,041)’: Do you mean frequency of mentions?

R: We have amended it for “abundance”. In Material and Methods it now reads (lines 164 to 169, page 5):

“The catalogued animals and gear were quantified for their absolute and relative abundances (the number of times a species or type of gear occurred in a given number of newspaper items) and richness (the number of different species and gear types in a given number of newspaper items) aggregated per decade. If species X was reported more than once in a single item, its abundance in the given item was counted as one.”

L234: ‘The number of items reporting marine animals using the keywords fishing (n = 269) and fish (n = 329) were significantly positively correlated with the number of newspapers (fishing r = 0.61; R2 = 0.37; p < 0.005; fish r = 0.52; R2 = 0.27; p < 0.02)’ Can you explain this more clearly? Are you referring to the total number of newspapers available (i.e. published online) in the region, or those identified through OCR searches as containing the words ‘fishing’ or ‘fish’? This is clarified somewhat by the legend in Fig 3 but it needs to be clear in the text, too.

R: Thanks, the sentence has been amended. Now it reads (lines 265 to 274, page 9):

“The number of items reporting marine animals using the keywords “pesca” (fishing) (n = 269) and "peixe" (fish) (n = 329) were significantly positively correlated with the number of newspapers for Santa Catarina available in the HDB archives (pesca r = 0.61; R2 = 0.37; p < 0.005; peixe r = 0.52; R2 = 0.27; p < 0.02), a pattern also observed by Gallardo et al.[10] (Figure 3A-B). Similarly, species richness and their abundances were also significantly positively correlated with the number of newspapers (species richness, r = 0.73; R2 = 0.53; p < 0.001, frequency, r = 0.68; R2 = 0.46; p < 0.001) in Santa Catarina (Figure 3C-D), indicating that, in absolute terms, our data is affected by the amount of digitised newspapers available in the HDB database.”

Fig. 3A and 3B – there are three colours in this panel figure, the black I assume correlates to the black text on the y axis descriptor (fig 3a), but there are also two red block colours in this figure and I am now sure how these align with B and the red text on the secondary y axis (#items). If these are aggregated per the legend description, why two red colours? Also, what does ‘items’ mean? Is this referring to individual articles within a newspaper edition?

R: There were only two variables, but the transparency in the plot created a third colour. The figure has been amended. “Items” refer to “articles, opinion letters, regulations and sanctions”, see lines 136 (page 5). We revised the figure caption for clarity. 

Fig. 3C-E – there seems to be very little difference between absolute species counts and species richness. It may be worth briefly repeating in the text (or legend description) the difference between these two measures, as it gets a bit confusing when examining the figure.

R: Their trends are similar but the numbers are different (please see their respective y-axes). However we noticed that we had switched the labels, which have now been corrected. 

In general, the way your results are written it reads as absolute species richness or species numbers, when actually it is richness or numbers reported in newspapers. I think this needs to be stated more clearly throughout the results. I.e., L256: By contrast, the richness and frequency of invertebrates REPORTED increased… L263: The WATL shows the prevalence of high trophic level species REPORTED over the last 180 years… L287: The diversity and frequency of fishing gear REPORTED did not change…

R: Yes, but it is implicit that these refer to reported species and gear in newspapers, not in absolute terms. However, for clarity we have revised the terminology through the text and in the Figure 3D-E. Absolute (relative) frequency has been replaced with absolute (relative) abundance (the amount of a species/gear in newspaper items aggregated per decade)

L466: you state the potential for newspapers to uncover species composition, better to clarify here that it is composition of fishery catch and/or landings? This of course reflects to some extent underlying ecological composition, but acknowledges the influences of social (in terms of what is reported) and market forces and technological trends (in terms of what is caught) on newspaper reporting.

R: The text has been revised to address this particular comment and now it reads (lines 509 to 515, page 16):

“However, contrary to other records such as official landing statistics, species composition in newspapers reflect distinct socio-ecological contexts, spanning from production to consumption systems, as well as considerations on the cultural and ecological values of some species, which are subject to evolving political, ideological and economic agendas, notably in social contexts with high levels of illiteracy as was the case in the late 19th and early 20th centuries in Brazil[12–14].”

L473: misspelling of ‘unsustainable’.

R: Thanks! 

L482: The relevance of this paragraph on LEK to the wider paper is unclear and it doesn’t tie in overly well. The historical context of conflicts you mention regarding LEK will undoubtedly also play out in popular media, it being a window into past cultures. But I’m not sure if you’re mentioning this to say that this is indeed the case with newspapers or to say that historical newspaper sources can mitigate this issue?

R: Thanks. The sentence has been revised for clarification. Our point here is about understanding how LEK are formed through time. The sentence now reads (lines 535 to 542, pages 17):

“Newspapers provide valuable insights into the changing nature of long-term human interaction with marine biodiversity, helping to track the origin and evolution of Traditional Ecological Knowledge. This knowledge plays a fundamental role in fisheries management[19], biological conservation[20,21] and coastal livelihoods[22–24], and newspapers reveal that it developed in a historical context of conflicts with institutions and political and economic actors, who often associated local fishers with backwardness, poverty, and destitution, and attributed the richness of their fishing practices with detrimental impacts and economic underdevelopment[10].”

L485. Your definition of LEK seems more akin to traditional ecological knowledge in terms of intergenerational transmission. While I am aware that TEK can be defined as a form of LEK, I think this needs to be clarified.

R: Yes, it is on ecological knowledge that is transmitted through generations. We have replaced LEK for TEK.

References

1. Freire KMF, Almeida Z da S de, Amador JRET, Aragão JA, Araújo AR da R, Ávila-da-Silva AO, et al. Reconstruction of Marine Commercial Landings for the Brazilian Industrial and Artisanal Fisheries From 1950 to 2015. Frontiers in Marine Science. 2021;8: 946.

2. Freire KMF, Aragão JAN, Araújo ARR, Ávila-da-Silva AO, Bispo MCS, Velasco G, et al. Reconstruction of catch statistics for Brazilian marine waters (1950-2010). University of British Columbia. Fisheries Centre; 2015. Report No.: 23 (4). doi:10.14288/1.0354313

3. Abdallah PR, Sumaila UR. An historical account of Brazilian public policy on fisheries subsidies. Mar Policy. 2007;31: 444–450.

4. Reis EG, D’Incao F. The present status of artisanal fisheries of extreme Southern Brazil: an effort towards community-based management. Ocean Coast Manag. 2000;43: 585–595.

5. D’Incao F, Valentini H, Rodrigues LF. Avaliação da pesca de camarões nas regiões Sudeste e Sul do Brasil (1965-1999). 2002. Available: http://repositorio.furg.br/handle/1/5716

6. Lopes PFM. Extracted and farmed shrimp fisheries in Brazil: economic, environmental and social consequences of exploitation. Environ Dev Sustainability. 2008;10: 639.

7. Valenti WC, Barros HP, Moraes-Valenti P, Bueno GW, Cavalli RO. Aquaculture in Brazil: past, present and future. Aquaculture Reports. 2021;19: 100611.

8. Souza SP, Begossi A. Whales, dolphins or fishes? The ethnotaxonomy of cetaceans in São Sebastião, Brazil. J Ethnobiol Ethnomed. 2007;3: 9.

9. Previero M, Minte-Vera CV, Moura RL de. Fisheries monitoring in Babel: fish ethnotaxonomy in a hotspot of common names. Neotrop Ichthyol. 2013;11: 467–476.

10. Sandoval Gallardo S, Fossile T, Herbst DF, Begossi A, Silva LG, Colonese AC. 150 years of anthropogenic impact on coastal and ocean ecosystems in Brazil revealed by historical newspapers. Ocean Coast Manag. 2021;209: 105662.

11. Caddy JF, Garibaldi L. Apparent changes in the trophic composition of world marine harvests: the perspective from the FAO capture database. Ocean Coast Manag. 2000;43: 615–655.

12. Pallares-Burke MLG. A imprensa periódica como uma empresa educativa no século XIX. Cadernos De Pesquisa. 2013;104: 144–161.

13. Oliveira RS. A relação entre a história e a imprensa, breve história da imprensa e as origens da imprensa no Brasil (1808-1930). Historiæ. 2011;2: 125–142.

14. Marchelli PS. As minorias alfabetizadas no final do período colonial e sua transição para o império: um estudo sobre a história social e educação no Brasil. EU. 2006;10: 187–200.

15. Soga M, Gaston KJ. Shifting baseline syndrome: causes, consequences, and implications. Front Ecol Environ. 2018;16: 222–230.

16. Fulton EA, Sainsbury K, Noranarttragoon P, Leadbitter D, Staples DJ, Porobic J, et al. Shifting baselines and deciding on the desirable form of multispecies maximum sustainable yield. ICES J Mar Sci. 2022 [cited 27 Sep 2022]. doi:10.1093/icesjms/fsac150

17. Campbell L, Gray N, Hazen E, Shackeroff J. Beyond Baselines: Rethinking Priorities for Ocean Conservation. Ecol Soc. 2009;14. doi:10.5751/ES-02774-140114

18. Freire KMF, Pauly D. RICHNESS OF COMMON NAMES OF BRAZILIAN MARINE FISHES AND ITS EFFECT ON CATCH STATISTICS. Journal of Ethnobiology. 2005;25: 279–296.

19. Tietze U. Technical and socio-economic characteristics of small-scale coastal fishing communities and opportunities for poverty alleviation and empowerment. FAO Fisheries and Aquaculture Circular; Rome N.o C. search.proquest.com; 2016. pp. 1–88. Available: https://search.proquest.com/openview/72e02c4d532ae5760e1481415b405b1b/1?pq-origsite=gscholar&cbl=237324

20. Reyes-García V, Fernández-Llamazares Á, Aumeeruddy-Thomas Y, Benyei P, Bussmann RW, Diamond SK, et al. Recognizing Indigenous peoples’ and local communities' rights and agency in the post-2020 Biodiversity Agenda. Ambio. 2022;51: 84–92.

21. Begossi A. Temporal Stability in Fishing Spots: Conservation and Co-Management in Brazilian Artisanal Coastal Fisheries. Ecol Soc. 2006;11. doi:10.5751/ES-01380-110105

22. Begossi A. Cultural and ecological resilience among caiçaras of the Atlantic Forest coast and caboclos of the Amazon. Linking social and ecological systems for resilience and sustainability The Beijer International Institute of Ecological Economics, Stockholm. 1998; 129–157.

23. Diegues ACS. Pescadores, camponeses e trabalhadores do mar. Editora Ática; 1983.

24. Diegues AC. A pesca construindo sociedades. NUPAUB-USP; 2004.

---

## [Editor Report · Decision Letter 1]

23 Mar 2023

180 years of marine animal diversity as perceived by public media in southern Brazil

PONE-D-22-32178R1

Dear Dr. Herbst,

We’re pleased to inform you that your manuscript has been judged scientifically suitable for publication and will be formally accepted for publication once it meets all outstanding technical requirements.

Kind regards,

Brian R. MacKenzie, Ph. D.

Academic Editor

PLOS ONE
---

## [Editor Report · Acceptance letter]

21 Jun 2023

PONE-D-22-32178R1 

180 years of marine animal diversity as perceived by public media in southern Brazil 

Dear Dr. Herbst:

I'm pleased to inform you that your manuscript has been deemed suitable for publication in PLOS ONE. Congratulations! Your manuscript is now with our production department. 

Kind regards, 

on behalf of

Prof. Dr. Brian R. MacKenzie 

Academic Editor

PLOS ONE